# Bladder Health Experiences, Perceptions and Knowledge of Sexual and Gender Minorities

**DOI:** 10.3390/ijerph16173170

**Published:** 2019-08-30

**Authors:** Cecilia T. Hardacker, Anna Baccellieri, Elizabeth R. Mueller, Linda Brubaker, Georgia Hutchins, Jory Luc Yimei Zhang, Jeni Hebert-Beirne

**Affiliations:** 1Howard Brown Health, Chicago, IL 60613, USA; 2Rush University College of Nursing, Chicago, IL 60612, USA; 3Community Health Sciences, School of Public Health, University of Illinois at Chicago, Chicago, IL 60612, USA; 4Departments of Obstetrics and Gynecology, Loyola University Stritch School of Medicine, Maywood, IL 60153, USA; 5Departments of Urology, Loyola University Stritch School of Medicine, Maywood, IL 60153, USA; 6Department of Obstetrics, Gynecology, and Reproductive Sciences, University of California San Diego, La Jolla, CA 92093, USA

**Keywords:** bladder health, qualitative inquiry, sexual and gender minority, community-based participatory research, focus group

## Abstract

While recent efforts have been made to understand the bladder health experiences, perceptions, and knowledge of cisgender adolescent females and women, virtually nothing is known about the bladder health experiences of people who identify as sexual and gender minorities (SGMs). A community-based participatory research approach using a focus group methodology to engage 36 adult participants who identify as SGM, including individuals who identify as gender non-conforming, queer, transgender (trans) men, or lesbian, in one of six focus group discussions on bladder health. Using directed content qualitative data analysis from the six unique focus groups, three interrelated themes were revealed: gender socialization of voiding behavior and toilet environment culture producing identity threats, and risks to gender affirmation; consequences of hetero-cis normative bathroom infrastructure necessitating adaptive voiding behaviors; and, physical and psychosocial consequences of chronic anxiety and fear are associated with voiding experiences. Insight on how SGMs navigate voiding behaviors, toilet experiences, and health care seeking is needed to assure that bladder health promotion activities are inclusive of this population’s needs.

## 1. Introduction

Bladder health is poorly understood, as the vast majority of bladder research has focused on the identification, treatment, and pathology of lower urinary tract symptoms (LUTS), which are highly prevalent in cisgender women. LUTS include issues relating to urine storage and emptying, urinary tract infections (UTIs, i.e., bladder infections), urinary incontinence, overactive bladder, incomplete emptying, bladder pain syndrome, and interstitial cystitis. In 2015, in response to the lack of LUTS prevention strategies, the Prevention of Lower Urinary Tract Symptoms (PLUS) Research Consortium was launched [1]. For its inaugural research activity, PLUS, a transdisciplinary, mixed methods research effort, implemented a national qualitative focus group (FG) study, Study of Habits, Attitudes, Realities, and Experiences (SHARE). SHARE enrolled 360 participants nationally, to examine the experiences of adolescent females and women with respect to their bladder health. Sexual and gender minority (SGM) individuals, assigned female at birth who identified their sexual orientation as lesbian or bisexual and people who identified their gender as gender non-conforming (GNC), queer, or transgender men, were not explicitly recruited for the SHARE study.

The bladder health experiences of SGM individuals are important to consider for two reasons. First, any national health equity promotion efforts designed without SGM insight could produce greater health inequities for SGM individuals and the SGM community [2]. Second, on-going, nationwide conversations on access to bathrooms for transgender and gender non-conforming or non-binary or genderqueer people in the United States may be informed by quality research on this topic. Without the benefit of research input, the state legislative sessions of 2015–2016 enacted country-wide increases in proposed legislation to prohibit access to public restrooms based on self-determined gender identity [3]; between 2013 and 2017, 24 states considered enacting laws to restrict public bathroom use based on one’s biological sex. By the end of 2018, eight states have enacted laws to protect students and employees’ access to bathrooms based on self-determined gender identity [4].

The Sexual and Gender Minority Opinions, Realities and Experience (SHARE-MORE) study, a community-based participatory research (CBPR) qualitative study was designed to contribute new knowledge about the experiences of certain SGM populations and to characterize their bladder-related experiences, perceptions, knowledge, and attitudes.

### Background

Little is known about healthy bladder habits, normal bladder function, or the meaning of a healthy bladder. What research does exist, focuses on cisgender women and adolescent females rather than SGMs, and focuses on the development and testing of behavioral and/or educational interventions [5,6,7,8,9]. The influence of social-ecology on bladder habits over the life course is also unknown. Insights into these aspects of human health and well-being, as well as applicable terminology for bladder function and health, are necessary for the development of effective health promotion interventions.

LUTS research has focused nearly exclusively on cisgender women, persons whose sex assigned at birth is female and whose current self-identity is female. Scant LUTS with SGMs research exists [10]. In most LUTS research, sexual/gender identification is restricted to a binary (male vs. female) paradigm. Sexual orientation information is not collected, reinforcing heteronormativity, the assumption that all participants identify as heterosexual. While it is likely that LUTS research may have unknowingly included SGM populations, SGMs are typically invisible to researchers if demographic surveys do not include sexual orientation and gender identity (SOGI) questions. Thus, cisgender individuals comprise the majority of participants in LUTS research intended to study “females”.

Well-documented, persistent health inequities exist in the SGM population, rooted in discrimination, societal stigma, and denial of civil and human rights [11,12]. Lesbians are less likely to seek preventive services for cancer [11,13]. Bisexual identified women experience higher rates of binge drinking, depression, anxiety, and sexual violence [14]. Transgender individuals experience a high prevalence of HIV/STDs, victimization, mental health issues, suicide, are less likely to have health insurance and may not seek screening for health issues related to their assigned sex [11]. And young transgender people report being bullied at school, feel unsafe on their way home from school, and were more likely to report violence victimization compared with their cisgender counterparts [15,16].

Recently, the issue of gender-segregated bathrooms gained national attention raising concerns for inequities in access to safe toileting spaces. Gendered toileting spaces discriminate against individuals who do not fall into the gender binary and reinforce existing gender norms [17]. Researchers found that avoidance of public bathrooms resulted in self-reported “weaker bladders” and that individuals intentionally navigate through certain areas or places depending on whether they will be able to access a safe bathroom [17]. Additionally, lack of gender-inclusive bathrooms in schools and work has resulted in transgender individuals dropping out of school or changing jobs [17]. Research indicates that those who had been denied access to a school bathroom or facility (in college) were 1.3 times as likely to have attempted suicide in some point in time, compared to those who had not [18].

The PLUS SHARE study is the first of its kind to explore experiences, perceptions, attitudes, and knowledge of bladder health. The SHARE-MORE study, the focus of this manuscript, is a complementary, stand-alone FG study that expands the goals of SHARE to include the experiences of certain SGMs groups.

## 2. Materials and Methods 

The institutional review boards of all partner institutions approved this research. This study used the CBPR approach and FG methodology to explore SGMs experiences, perceptions, and knowledge about bladder health within a social-ecological framework across the life course. CBPR is a participatory approach to research in which academic and community-partners engage in equitable partnership across the research process [19] and findings are grounded in lived experience [20,21]. One of the nation’s largest lesbian, gay, bisexual, transgender, and queer (LGBTQ) organizations and a federally qualified health center was the lead community partner bringing cultural expertise to the study and hosted the six FGs. Consistent with CBPR principles, funding and roles were shared across research and community partners. Qualitative methods like FGs produce contextual findings to explore complex phenomena, social processes, and unpack meanings people attribute to experiences [22].

Individuals with the following SGM identities were intentionally recruited for the study: those who were assigned female gender at birth and now identify as GNC, queer, transgender men, lesbian or bisexual. The following SGM identities refer to: lesbian, romantic and/or sexual attraction to people of the same gender identity; bisexual, romantic and/or sexual attraction to people of more than one gender; GNC who view their gender identity as other than strictly man or woman, but of many possible gender identities; queer, umbrella term for individuals with fluid gender identities and sexual orientations, other than heterosexual or cisgender; transgender men, individuals whose male gender identity does not align with their assigned female gender at birth [23]. Excluded from participation were individuals not comfortable speaking and reading English; outside the age criteria; pregnant; identified as heterosexual and cisgender women; born with or retain a prostate, indicating people who were cisgender men, intersex individuals, and transgender women.

The guide included 5 sections (18 questions with probes): healthy bladder, knowledge acquisition, LUTS and care-seeking, terminology, and public health messaging. These aligned with the PLUS Conceptual Framework that emphasized social-ecological levels of influence across the life course [24], comprising individual/biology body, individual mind/behavior, interpersonal, institutions and organizations, society and community. Examples include: What are your ideas about what it means to have a healthy bladder? How does it feel to have a healthy bladder? What are some of the things that help people to have a healthy bladder/unhealthy bladder? Anything about our environment at home (school, work, and in public)?

Moderators are the research instrument in FG methodology. The lead moderator (6th author) was selected, in part, for their strong identification with and advocacy for the SGM community. 

Moderators were trained by both academic and community partners and grounded the SHARE conceptual framework, qualitative research principles and best practices, and the study protocol to increase the fidelity of the FG guide and trustworthiness of data. The moderators participated in the FG training videos and discussion board posts housed on the online training platform, as well as the Audio and Listen and Reflection of mock FGs and conducted their own mock FG. In addition to becoming well versed in the research aims and constructs framing the FG guide, it was equally important for the FG moderators (and broader research team) to relate to participants in culturally relevant ways. The community partner and lead moderator provided ongoing training to research team colleagues on common gender identity and sexual orientation terminology and definitions; additional relevant language (i.e., passing), particularly as it relates to transitioning processes, and as it relates to toileting, phalloplasty’s, “bottom surgery”, changes from hormones, toileting positions, and stand-to-pee devices; and relevant literature, including experiences of SGM individuals and the socio-political climate of gender identity and public restrooms.

Purposive recruitment of participants was led by the partner organization who also hosted the FGs. Recruitment strategies included social media, community recruitment, word of mouth, and flyers.

Participants signed the informed consent form. Research staff included the moderator, assistant moderator, and field note taker. Each participant created a pseudonym to reduce the risks of a loss of privacy. The moderator introduced themselves and followed the guide and probed as necessary. At the end of the 90 min audio-recorded FG, participants completed demographics, medical history, bladder health, and toileting behaviors questionnaires. Once completed, participants received a $50 gift card for their time. The study flow is presented in Figure 1. 

The validated LUTS survey assessed the frequency and bother [25,26]. Socio-demographics variables included age, race/ethnicity, and SOGI data. Social determinants of health data collected included education, socio-economic status, and occupation. The validated 26-item Toileting Behavior-Women’s Elimination Behaviors Scale (TB-WEB) [27,28] was used to measure voiding behaviors in public and private settings.

### Approach to Analysis

The participatory analysis and interpretation team assured SGM expertise input throughout the process. The team adapted the SHARE analytic approach, involving directed content analysis (DCA), data interpretation through data immersion, team dialogue, and adherence to standard qualitative data management and data analysis approaches [29].

Participatory analysis and interpretation occurred at multiple stages, including debriefing sessions; transcript cleaning, memoing and coding processes; team meetings; and member-checking. The team used standard qualitative data analysis techniques [30,31,32] and DeDoose® qualitative data analysis software. This process offered a standard and participatory approach to test and adapt the SHARE codebook to best represent the SHARE-MORE data. The team added new codes and expanded code definitions. Two coders trained in the codebook and in DCA first read through and memoed in response to two queries: (1) What are you hearing? and (2) How does it help us understand our primary aims? The team met weekly to resolve coding variations and further refine the codebook. Each code specified an operational definition with inclusion and exclusion criteria. Variations within codes generated sub-codes. The team examined code frequency, code co-occurrence, and code by primary document to identify patterns. Analysis of patterns and associations across codes and coded text segments led to the identification of emerging insights until saturation was met. The team tracked code excerpts based on these insights and selected quotes exemplifying the theme for inclusion in the manuscript. The team engaged in two 1-hour member checking sessions that included two original participants to confirm the trustworthiness of findings and discuss emergent themes. 

## 3. Results

### 3.1. Overview

Forty-two participants completed the phone interview screener. Five participants either canceled or did not attend. Six FGs were organized by SGM identity and conducted between May and August 2018. Table 1 presents the group characteristics of the 36 participants (5–9 participants per group). 

### 3.2. Participant Characteristics

Table 2 displays participant socio-demographic characteristics. There were 12 participants (33.3%) who identified as female/woman, 7 (19.4%) as trans male/trans men, 8 (22.2%) as genderqueer/GNC, and 9 (25%) identified in a different way. The majority of participants (22 (61.1%)) identified as queer, five (13.9%) selected bisexual, and four (11.1%) selected lesbian. One participant identified as heterosexual/straight. The majority (75%) of participants identified as White, and two participants identified as having Hispanic/Latino/Spanish origin. Table 3 presents the social determinants of health characteristics. Nearly 40% of participants’ highest degree completed was a bachelor’s degree (n = 14) and 25% of participants (n = 9) highest degree was a master’s degree. Financial concerns were common: many reported not having enough, seven (19.4%), or just having enough, 17 (47.2%), money to make ends meet at the end of the month. 

Table 4 presents the key LUTS characteristics. Most (n = 25 (69.4%)) participants reported LUTS, including leaking urine (n = 8 (22.2%)), leaking in connection to sudden need to rush to urinate (n = 5 (13.9%)), and in connection with laughing sneezing or coughing (n = 5 (13.9%)). Nearly one in five (n = 7 (19.4%)), participants reported leaking urine after voiding. All trans men (age 18–25) reported LUTS. 

### 3.3. Emergent Themes

Three major themes emerged regarding the experience, perceptions, attitudes, and knowledge about bladder health. First, participants illustrated how gender socialization of voiding behavior/bathroom culture and toilet environments produces threats to identity and affirmation. Second, they perceive that their adaptive behaviors are an unhealthy response to the experiences of cis-heteronormative bathroom infrastructure. Lastly, participants disclosed physical and psychosocial consequences of chronic anxiety/fear associated with voiding experiences. Thematic evidence, as represented by quotes from focus group participants, follows each theme indicating the corresponding FG. 

#### 3.3.1. Gender Socialization of Voiding Behavior/Bathroom Culture and Toilet Environments Producing Threats to Identity and Affirmation

##### Bathroom Culture

Groups with transgender men, GNC, and queer participants described a gendered bathroom culture, with gendered norms and expectations around bathroom use. The term gendered was frequently used to describe toileting as a male (for transgender men), or in comparison to males, or the differences between male and female toileting. It also included bathroom choice as gender validation; by deciding to use the bathroom of their choice was supportive of their personhood. Male bathroom culture was experienced and socially understood as private and personal with transactional expectations to just do your business, get in and out, do not look, at each other or at each other’s penis. The experience in the bathroom is restricted to emptying bladder or bowels and other activities e.g., fixing hair in mirror, talking, socializing is out of the norm and unexpected. Participants described that men do not talk in bathrooms and do not go in group or pairs. This was a sharp contrast to female bathroom culture, experienced and perceived as social, space for solidarity building. It is a space for staging and back staging to prepare to present publicly [33]. This culture encourages going in groups, socializing, taking your time, and using the time as a retreat or escape from what is outside the bathroom. This gendered contrast was evidenced by a contribution from a focus group participant who identified as queer,

*And like, queers have intimacy with each other because we’re part of the same group, kind of speak the same language, have the same general like ideas of things. And that’s why that’s comfortable for us to do that. And like I don’t know about like cis dudes, though, because like they’ll talk about their bodily functions all day long. ‘Blah, blah, blah, blah, blah’. But then when it gets to the bathroom time, they’re like, ‘I can’t look at you. Like I can’t make eye contact. I can’t --- I’m gonna try not to listen to the sounds that you’re making. I assume you are peeing, but I don’t want to know’*.

*Like just stare straight ahead because if I look something bad will happen. ‘It’s just really weird’. And it’s like, ‘Okay. Like, everybody pees,’ but I guess if you look at someone peeing, it makes you gay. And all of a sudden, you’ll be gay because you saw someone peeing in a public bathroom. [Laughter] Which, if that did cause you to become gay, you were probably already gay but you didn’t know it yet. ‘Calm down’ (Queer, 26–44)*.

In a male bathroom, trans men avoid engaging in female normative behavior because it could increase the risk that they would not pass as male.

*I might even be bringing this upon me, but I think this is part of my self-care ritual. So I really love my hair. And I like to do my --- like if I’m in the bathroom I always have to like check up on my hair. And it’s --- I always forget sometimes that there’s, you know, like if I’m in a public restroom*.

*There’s small things that I’m just accustomed to doing, and I don’t want to change because then I feel like I’m giving in to this like expectation of performing cis masculinity. So I’m like ‘Fuck it’ … I’m gonna do me.’ You know, slowly subverting, you know, gender binary, one mirror at a time. And, umm, there are small things that happen that I’m just not aware of like social cues. So I use the hair thing, really, as symbolically to represent other small mannerisms that I have that I’m just not aware of in bathrooms, in like public bathrooms shared by cis men that I think also like impact my experience being in bathrooms. Cuz’ then it turns into this very anxious experience of how quickly can I get into the bathroom and get out, which historically has never been the experience. Like ---So I think, yeah, there are just customs surrounding bathrooms that I learn the hard way. I’m like, ‘Oh my god. Like, I made eye contact. Didn’t know I was supposed to do that.’ You know, just like whatever it is. I’m always learning something new in like a very violent and like disciplinary kind of way. (Trans men, 18–25)*.

Going into male bathrooms alone can be perceived as a real danger. Participants described themselves or others being assaulted because they did not pass as male and were presumably perceived as a threat to cisgender men using the bathroom. Transgender men, GNC, or queer individuals often feel that they need a companion to accompany them to the bathroom to reduce the risk for violence. However, going with others breaks the social norm of going alone, thus increasing threat to identity, described as failure of passing as a male. 

Gendered bathrooms are also spaces where one can seek affirmation for one’ gender identity and, in some ways, the bathroom becomes the ultimate test. 

*But while I was struggling with it, it was just like, umm, like th --- the things one would do for like even a little bit of validation. Like instead of using like the gender-neutral bathroom, like to just use the men’s bathroom even if it was like really risky*.

*Like I would go to my locker, like find a binder. Cuz’ I couldn’t wear it all day because it hurt. And, umm, I’d put up my hair in like a hat and then a hood. And then I’d take like dark makeup and like shadow my face a certain way. And then I’d like go into the bathroom for like a minute and get out. And it’s like, all that for like that. It was a lot (GNC 18–25)*.

##### Early Childhood Experiences

Participants recalled early childhood experiences that demonstrated ways in which gender was socially constructed and currently impacts their bladder health and behavior. One participant described this experience,
One of the first like memories that I have when it comes to like my own gender identity, is like getting in trouble for peeing standing up in daycare. Umm, and like, that really fucked me up, big time. Umm, and like, I like -- after that, had like some big problems with voiding, just in general. And my mom had to read me this like ‘Everybody Poops’ book like a hundred times a day to like get me to go. Umm, cuz’ I was just like, ‘I’m just not gonna go to the bathroom if I can’t go the way I want to go’. (GNC, 18–25)

Another participant shared a similar experience,
Like I’m getting the vibe that this is like a similar story, like just a similar experience for people. Cuz’ like I was known as this kid who would like insist that they can -- like the parent would insist that ‘I’m like potty-trained’ and stuff. But I would constantly like end up having like accidents in this like private bathroom that we had in the kindergarten. And the thing that they didn’t know was that whenever I was at school, I would just constantly try to pee standing up. And it would like never work out. So the teacher always had to be calling my mom to bring me new clothes. And like I just kinda just got this rep with the teacher where she was like, ‘This kid doesn’t like know how to like function in this age yet.’ But I don’t know, it was just like this – I – I don’t think, this thing that like ‘kids are not addressed somehow.’ I don’t know. So, it was probably like my own confusion. I thought a certain part of the self was a dick so I was just like, ‘This should work. Why isn’t it working?’ (GNC, 18–25)

##### Voiding Positions: Stand to Pee (STP)

Trans men, GNC, and lesbian participants discussed the benefits and challenges using stand to pee (STP) and Go-Girl devices to accommodate the expediency of voiding while standing. The STP devices marketed primarily to transgender men are costly and resemble an anatomical phallus, as opposed to STPs for cis-populations, which are nondescript and inexpensive. These utilitarian devices for cisgender women are gendered, colored pink or blue and named “white people” names, such as Jack or Jane. One participant described, “me and my friends, we used to try to – to practice standing up peeing. We -- we did that. No STP. I didn’t even know about that being. But we used to actually stand up and, you know, practice doing it” (Trans men, 26 +).

#### 3.3.2. Hetero-Cis-Normative Bathroom Infrastructure Necessitates Adapted Behaviors

##### Built Environment

Participants describe the toileting environment as generally built from a hetero-cis-normative perspective; thus, they needed to engage in significant adaptive behaviors to be able to void in the toileting environment as they preferred. Participants voiced the concern that stall doors often do not provide privacy in either the gender-neutral bathrooms or in male bathrooms. There are few stalls in cis-male bathrooms and stalls can be built so that there is a loss to privacy through the cracks in the doors. Even toileting environments in SGM friendly spaces such as bars, a setting that often emerges as an important space for SGMs given its role historically as one of a few spaces SGMs can safely be themselves, are not private. Bar management may remove doors, latches or make sure doors do not close in gay bars to discourage sexual behavior in stalls (bathrooms in gay bars have historically been a space where sex occurs given the lack of other safe spaces in a heteronormative society) eliminating safe spaces for trans men, GNC, or queer individuals to void who seek privacy in their voiding behaviors. A participant disclosed, *“Or, you know, if I am standing in a stall and the door doesn’t close, like sometimes that’s kind of an invitation. So, it’s better if I can just find a door that closes. Um, that way, everyone knows, like, this is not an invite for an interaction” (Trans men, 26 +).*

The lack of privacy leads to a variety of adaptive behaviors. Participants described excessive holding, leaving the space to void at home and/or avoiding public spaces to void. Participants often do not use the bathrooms if it is occupied; instead, they hold their pee, monitoring the bathroom until it is empty, or walk long distances with a full bladder to find another. Some participants use the bathrooms and expressed their anxieties while in the stall, further discussed in the next emergent theme. Other participants described attempting to go and then try to make themselves go faster if they hear someone coming.

*Uhmm, so I’ve ended up, like especially when I was pre-teen, I didn’t feel like I was passing super well, and I was way more nervous to use restrooms, I would walk all the way home and then come back to a bar instead of using the bathroom there. So I would hold it a lot longer than I needed (Trans men, 26–44)*.

A conversation from the Transgender men, 18–25 FG, offers support around voiding norms. It also conveys a heightened awareness of being monitored by others and heightened anxiety around meeting expectations. (Note: pseudonyms are used below in lieu of participants names).

*Austin: Umm, I actually have four roommates, so it’s not really like I’m ever really totally alone, but it’s great not to have to be around a bunch of cis people who like assume that I’m poopin’. [Laughs] All the time. I don’t know. I can just go to the bathroom and not have to worry like how my pee sounds. Like, ‘Does my pee sound masculine enough?’ It sounds ridiculous. But like, and also no one is looking under the stall. I don’t know*.


*Moderator: But that’s real. Do – do you think that there is this concept of what "masculine pee" sounds like?*



*Austin: In my head, yes. (Laughs)*



*Moderator: So what does ‘masculine pee’ sound like?*



*Austin: I don’t know. Like I feel like it’s more of a stream and less of a spray. (Laughter)*



*Xan: It’s like the sound coming from like a different angle. And also --- this is Xan --- and looking at the direction of the shoes like under the stall. It’s like --- (sighs)*


#### 3.3.3. Physical and Psychosocial Consequences of Chronic Anxiety/Fear Associated with Voiding Experiences

Participants clearly conveyed that the risks and consequences of not passing or of being discovered as “other” are real. This clearly leads to heightened anxiety, intense mapping and extreme/excessive adaptive behaviors. The bathroom, which for some is a place to define oneself and regroup, is also a place in which violence and harm may occur, especially based on one’s identity.


*It’s happens to be like the place that’s most violent and anxiety-producing also happens to be sometimes the safest space, cuz’ I’m like literally in a stall (Trans men, 18 +).*


Failures in presenting oneself have significant mental and physical consequences, harming potentially one’s sense of personhood, mental health, and physical self either through excessive routine holding or physical violence by others. Participants disclosed routinely the need to find a place where they feel safe.

##### Feeling Like One is Constantly Being Read

Participants described constant monitoring of their environment, social ecology and selves; they sense that they are constantly being read or evaluated. 


*And I’m kind of, you know, I’m – I’m aware that my appearance can be a bit out there at times, so people are gonna look regardless. So I just – you know if I have to use the bathroom, I don’t care if I walk into the women’s or the men’s bathroom, because I need to use the fucking bathroom, in a way. Like I don’t care if somebody’s gonna react. You know?*



*But, yeah, as far as, like, going into the men’s restroom, the urinals, I don’t do the urinals. I just go straight (Laughs) you know --- straight into the stall, do my thing. And, yeah, there are times where it does hit me. I’m, like, ‘oh, I’m getting ready to sit down and like pee. You know, maybe somebody is kinda, like, looking at my feet, seeing that they’re not facing the other direction or something like that.’*



*But I’m, like, ‘fuck it, let em’ think that I got a small dick or something’. (Laughter) You know, and that’s why I’m sittin’ down. I’m just, like, you know, I --- I don’t know. I just, like, I know that these are like very, very real things. Um, you know, just for my own personal experience, I --- I don’t --- I think there’s so much more I could be stressed out about. So I just like try not to like stress myself out with things like that (Trans men, 25 +)*


Many describe anticipating reactions from others and planning accordingly.


*And so I often use the men’s room. And, umm, I always sort of like -- when I’m going to leave, I like prepare myself. I’m like, ‘Okay. What if there’s a dude like waiting?’ And part of me is like what you were saying earlier of sort of like a fear of like men, and just like ‘Am I gonna get beat up? Am I gonna – like what’s gonna happen?’*



*And the other half of me is like sort of like excited to like fuck with their minds [group laughter], to see the look on their face when they’re like, ‘A not-dude just came out of the dude’s room. Like what does this mean about me and my manhood?’ So it’s like -- I kind of enjoy it, but also a little bit -- it can be scary (GNC 18–25)*


##### Acceptable Terminology 

Our study revealed that bladder terminology is contextually and situationally specific with more formal language suited to professional settings and more informal language appropriate in settings with family and friends often with a sense of humor.


*In a more professional setting, it would be like, "I have to go to the bathroom," or nothing. Whereas with friends and family, it’s like, "I have to pee." And usually it’s, "I have to pee again." [Laughter]. Or I say, "I have to pee," and they’re like, "Again?" (Group laughter) "Yes..." (Lesbian 45–64).*


Some suggested crude language such as *"I need to take a dump"* or *“I need to take a whiz"* or *“I have to take a piss"* with people in one’s social networks. Many emphasized the acceptability of informal language, *“I don’t think ‘pee’ is crass. I think that’s actually the accepted word these days” (Lesbian 45–64).* Most participants found “I have to pee” is the most common, accepted terminology. 

However, many discussed gender differences in terminology with masculine terminology being more *“bawdy”* such as saying, *"I have to piss like a racehorse,"* and just like run around because they’re like, *"I can, I can pee wherever I want. I’ve got a firehose" (Queer 26–44).* Whereas one participant described a contrasting experience:


*I also kinda feel, contrasting, that there’s like a really like feminine like quality to having to go pee a lot, like it’s like socialized to be a very girly, like girly thing, whatever that means. Like me and my friends who are femmes or women, we always, "Oh, let’s go pee. I gotta go pee." And we pee in front of each other or like all go to the bathroom together, and it’s not like gross or anything (Queer, 26–44).*


Overall participants emphasized the importance of language that is inclusive of a variety of experiences and identities and respect for the whole person. Participants discussed a desire for a more compassionate society as a whole that understands the level of anxiety associated with bladder norms and bathroom behavior and works to create a more flexible, tolerant environment.

## 4. Discussion

We found that SGM FG participants described that bladder health is influenced by gendered bathroom and voiding norms and expectations, cisgender-built toileting environments, and the consequences of anxiety associated with bladder-related behaviors and experiences. Our findings emphasize important issues associated with accessing bathrooms, navigation of bathrooms, and bathroom behaviors, adaptive behaviors, and monitoring. These issues emerged in SHARE findings but emerged with greater intensity in SHARE-MORE; the consequences were also much more significant for the studied SGM participants, particularly the transmen and GNC participants as demonstrated by the thematic evidence. One observation from research with cisgender women reveals that they are focused on urinary leakage and the shame and embarrassment of being discovered “wet” which can affect their sense of self and social image. While the SGM groups disclosed having similar experiences and feelings, they were secondary to the physical threat of being discovered (as transgender men) and their lack of safety in certain bathroom spaces. Some of the insights about bathroom access have emerged in other recent research with SGMs. Bathroom access was one of the most frequent concerns discussed by 18 transgender individuals in another study, and these participants reported feeling fearful when bathroom stalls did not lock, and they desired safe, clean, and lockable spaces [34]. A study on minority stress found that transgender and GNC individuals described feelings of discomfort, stress, anxiety, and anxiety coupled with anger in relation to public bathroom use [35]. An additional study found that participants were aware that holding the bladder is bad for health, but the other option, using a public bathroom, is riskier for transgender and GNC people [17].

In building a national prevention research agenda, the salient issues of SGMs, especially the heightened anxiety experienced, are important to consider so as not to exclude SGM individuals from national health promotion opportunities. For SGM populations, public health education is needed on the consequences of urine holding and the impact of anxiety on storage and emptying.

As demonstrated in Table 1 and Table 2, several participants selected different gender identities on the socio-demographic characteristics questionnaire, than they did on the phone interview screener. This likely is attributed to (a) evolving terminology and definitions (i.e., moving from GNC toward genderqueer) and (b) exemplifies the tensions of societal norms to label and categorize gender identity and sexual orientation as opposed to experiential, complex understandings of subjective, temporal, fluid, and possible spatial nature.

The important work started in this study should not be assumed to represent the perspectives of all SGM individuals. The transmen and GNC (nonbinary and genderqueer) focus groups presented the most significant thematic evidence relative to the lesbian and bisexual focus groups perhaps due to the greater degree to which they are marginalized from society based on their identification outside of the gender binary. Our research may over-represent issues of young SGM because three of the groups we hosted were young, 18 to 25-year-olds. These themes may also be reflective of a particular with socio-political climate that is hostile to SGM issues [36,37]. Since 2016, several anti LGBTQ policies have been put in place, including the removal of SOGI questions from national aging and disability surveys [38,39] and other health policies have been proposed that would have a negative impact on SGM, especially SGM people of color [37]. Namely, this socio-political environment may have increased generalized anxiety for this group.

Older lesbians presented the greatest recruitment challenge. Recruiters attempted to utilize all modes of recruitment and ultimately relied on word of mouth and personal connections to known people in the community. Difficulty with the recruitment of older lesbian-identified women for research is known due to previous harmful experiences within the healthcare system, mistrust of researchers, and fear of disclosure [40].

SHARE-MORE participants seemed to make more meaningful connections between negative childhood experiences and current bladder experiences. It is possible this awareness is due to the experience of marginalization of this population due to sexual and/or gender minority status. For example, SGM participants described that they are often in situations in which they have to advocate for themselves. The health care system emerged as an example of a system that is largely seen as a hostile, inaccessible, sometimes irrelevant to this population. Participants described the need to frequently advocate for oneself by teaching practitioners about themselves and broader SGM health issues.

## 5. Conclusions

The negative experiences with voiding behavior and bathroom spaces among SGM population are significant with potentially severe consequences. The insights from this study will inform a better health promotion plan for SGM populations which will include promoting policies that avoid restricted bathroom access. Additional research is needed to gain insight into bladder health experiences of other SGM individuals who were not included in this study.

## Figures and Tables

**Figure 1 ijerph-16-03170-f001:**
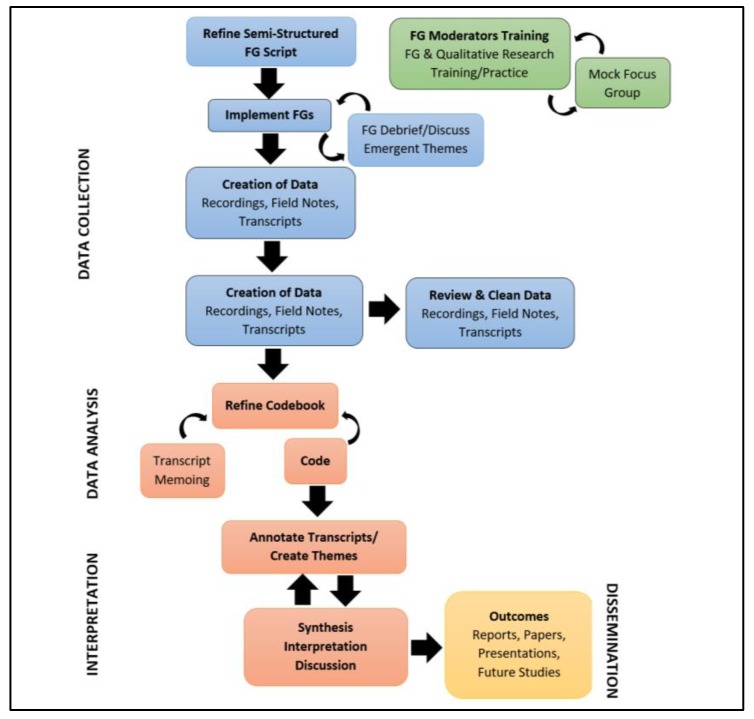
Study flow diagram.

**Table 1 ijerph-16-03170-t001:** Focus group characteristics.

SGM Female Identity Assigned at Birth (FAAB)	Age Group	Number of Participants
Lesbian	45–64	5
Bisexual	18–25	5
Gender Non-conforming (GNC)	18–25	6
Queer	26–44	9
Trans men	18–25	5
Trans men	26–44	6
Total		36

**Table 2 ijerph-16-03170-t002:** Focus group participants socio-demographic characteristics, number (percentage).

Participant Characteristics	OVERALL (n = 36)	QUEER 26–44 (n = 9)	TRANS MEN 18–25 (n = 5)	TRANS MEN 26–44 (n = 6)	GNC 18–25 (n = 6)	BISEXUALS 18–25 (n = 5)	LESBIANS 45–64 (n = 5)
Age (mean ± SD)	m = 30	28.44 ± 2.83	22 ± 2.68	33 ± 4.86	23.33 ± 1.75	22 ± 1.92	53 ± 6.42
Gender Identity							
Female/Woman	12 (33.3)	5 (55.6)	0	0	0	3 (60)	4 (80)
Trans male/Trans man	7 (19.5)	0	1 (20)	4 (66.7)	2 (33.3)	0	0
Genderqueer/GNC	8 (22.2)	3 (33.3)	1 (20)	0	3 (50.0)	1 (20)	0
Identify in a different way	9 (25)	1 (11.1)	3 (60)	2 (33.3)	1 (16.7)	1 (20)	1 (20)
Romantic/sexual attraction to others							
Heterosexual	1 (2.8)	0	0	1 (16.7)	0	0	0
Lesbian	4 (11.1)	0	0	0	0	0	4 (80)
Gay	2 (5.6)	0	0	2 (33.3)	0	0	0
Bisexual	5 (13.9)	0	0	0	1 (16.6)	3 (60)	1 (20)
Queer	22 (61)	9 (100)	5 (100)	3 (50)	4 (66.7)	1 (20)	0
Something else	2 (5.6)	0	0	0	1 (16.7)	1 (20)	0
Race (Check all that apply)							
American Indian/Alaska Native	1 (2.8)	0	0	0	0	0	1 (20)
Asian	3 (8.3)	0	0	2 (33.3)	1 (16.7)	0	0
Black/African American	7 (19.4)	0	1	2 (33.3)	2 (33.3)	2 (40)	0
White/Caucasian	27 (75)	9 (100)	4 (80.0)	3 (50.0)	4 (66.7)	2 (40)	5 (100)
Some other	3 (8.3)	2 (2.22)	0	0	0	1 (20)	0
Ethnicity							
Not Hispanic, Latino, or Spanish	34 (94.4)	5 (77.8)	5 (100)	6 (100)	6 (100)	5 (100)	5 (100)
Hispanic, Latino, or Spanish Origin	2 (5.6)	2 (22.2)	0	0	0	0	0
Primary Language Spoken at Home (Check all that apply)							
English	35 (97.2)	9 (100)	5 (100)	6 (100)	5 (83.3)	5 (100)	5 (100)
Another Language	1 (2.8)	0	0	0	1 (16.7)	0	0
Marital Status							
Now Married	6 (16.6)	2 (22.2)	0	0	1 (16.7)	0	3 (60)
Divorced	1 (2.8)	1 (11.1)	0	0	0	0	0
Never Married	27 (75)	6 (66.7)	5 (100)	5 (83.3)	4 (66.6)	5 (100)	2 (40)
Did not answer	2 (5.6)	0	0	1 (16.7)	1 (16.7)	0	0
Single Status							
Living with a partner	9 (30)	3 (33.3)	1 (20.0)	0	2 (40)	1 (20)	2 (40)
Committed relationship, not living together	3 (10)	1 (11.1)	1 (20.0)	0	0	1 (20)	0
Seriously dating; not committed	3 (10)	1 (11.1)	1 (20.0)	0	0	1 (20)	0
Casually dating	5 (16.7)	1 (11.1)	1 (20.0)	1 (16.7)	1 (20)	1 (20)	0
Not dating/or with a partner	9 (30)	1 (11.1)	1 (20.0)	4 (66.6)	2 (40)	1 (20)	0
Did not answer	1 (3.3)	0	0	1 (16.7)	0	0	0
Totals may not equal 100% due to missing data or NA							

**Table 3 ijerph-16-03170-t003:** Social determinants of health, number (percentage).

Participant Characteristics	OVERALL (n = 36)	QUEER 26–44 (n = 9)	TRANS MEN 18–25 (n = 5)	TRANS MEN 26–44 (n = 6)	GNC 18–25 (n = 6)	BISEXUALS 18–25 (n = 5)	LESBIANS 45–64 (n = 5)
Household Income							
Less than $10,000	6 (16.7%)	1 (11.1%)	0	1 (16.7%)	2 (33.2%)	2 (40%)	0
$10,000–$24,999	11 (30.6)	4 (44.5)	5 (100)	1 (16.7)	1 (16.7)	0	0
$25,000–$49,999	9 (25)	3 (33.3)	0	4 (66.6)	1 (16.7)	0	1 (20)
$50,000–$74,999	2 (5.5)	0	0	0	1 (16.7)	1 (20)	0
$75,000–$99,999	3 (8.3)	0	0	0	0	1 (20)	2 (40)
$100,000–$124,999	2 (5.5)	0	0	0	1 (16.7)	1 (20)	0
$125,000 or more	1 (2.8)	0	0	0	0	0	1 (20)
Don’t know/not sure	1 (2.8)	1 (11.1)	0	0	0	0	0
Participant did not answer	1 (2.8)	0	0	0	0	0	1 (20)
At the end of the month do you generally							
Not have enough money to make ends meet	7 (19.4)	1 (11.1)	0	2 (33.3)	2 (33.3)	2 (40)	0
Just have enough money to make ends meet	17 (47.3)	6 (66.7)	5 (100)	2 (33.3)	2 (33.3)	1 (20)	1 (20)
Have some money left over	8 (22.2)	2 (22.2)	0	0	2 (33.3)	2 (40)	2 (40)
Have more than enough money left over	4 (11.1)	0	0	2 (33.3)	0	0	2 (40)
Employment (check all that apply)							
Homemaker	4 (11.1)	2 (22.2)	0	1 (16.7)	1 (16.7)	0	0
Student-part time	3 (8.3)	0	1 (20)	2 (33.3)	0	0	0
Student-full time	10 (27.8)	0	1 (20)	1 (16.7)	4 (66.7)	4 (80)	0
Unable to work	3 (8.3)	1 (11.1)	0	0	0	2 (40)	0
Out of work/Unemployed	13 (36.1)	4 (44.4)	0	3 (50.0)	3 (50)	3 (60)	0
Working one or more jobs	33 (91.7)	7 (77.8)	5 (100)	5 (83.3)	6 (100)	5 (100)	5 (100)
How many hours a week do you work? (mean ± SD)	33.65 ± 12.95	31.71 ± 21.89	28.60 ± 8.05	38 ± 3.46	32.50 ± 15.73	30 ± 7.07	44.50 ± 5.26
Education							
Regular high school diploma	2 (5.6)	1 (11.1%)	1 (20)	0	0	0	0
Some college credit; no degree	7 (19.4)	0	2 (40)	1 (16.7)	3 (50.0)	1 (20)	0
Associate’s degree (AA/AS)	2 (5.6)	0	0	1 (16.7)	0	0	1 (20)
Bachelor’s degree (BA/BS)	14 (38.8)	5 (55.6)	2 (40)	3 (50)	2 (33.3)	2 (40)	1 (20)
Master’s degree (MA, MS, etc.)	9 (25)	3 (33.3)	0	0	0	2 (40)	2 (40)
Professional degree (MD, DDS, etc.)	1 (2.8)	0	0	0	0	0	1 (20)
Doctorate degree (PhD, EDd, etc)	1 (2.8)	0	0	1 (16.7)	0	0	0
Homeless in the past year?							
No	35 (97.2)	9 (100)	5 (100)	6 (100)	5 (83.3)	5 (100)	5 (100)
Yes	1 (2.8)	0	0	0	1 (16.7)	0	0
Stayed in a Shelter, even for a night?							
No	35 (97.2)	9 (100)	5 (100)	6 (100)	5 (83.3)	5 (100)	5 (100)
Yes	1 (2.8)	0	0	0	1 (16.7)	0	0
Been in transitional housing, even for a night?							
No	33 (91.7)	9 (100)	4 (80)	5 (83.3)	5 (83.3)	5 (100)	5 (100)
Yes	3 (8.3)	0	1 (20)	1 (16.7)	1 (16.7)	0	0
Where do you live?							
One-family house detached from any other house	6 (16.7)	0	1 (20)	0	1 (16.7)	2 (40)	2 (40)
A building with 2–4 apartments	12 (33.3)	4 (44.5)	2 (40)	3 (50)	1 (16.7)	1 (20)	1 (20)
A building with 5–19 apartments	9 (25)	3 (33.3)	0	2 (33.3)	2 (33.3)	1 (20)	1 (20)
A building with 20 + apartments	9 (25)	2 (22.2)	2 (40)	1 (16.7)	2 (33.3)	1 (20)	1 (20)
Currently have health insurance?							
Yes	30 (83.3)	6 (66.5)	4 (80)	5 (83.3)	5 (83.3)	5 (100)	5 (100)
No	6 (16.7)	3 (33.5)	1 (20)	1 (16.7)	1 (16.7)	0	0
Totals may not equal 100% due to missing data or NA							

**Table 4 ijerph-16-03170-t004:** Key lower urinary tract symptoms.

LUTS Characteristics	OVERALL (n = 36)	QUEER 26–44 (n = 9)	TRANS MEN 18–25 (n = 5)	TRANS MEN 26–44 (n = 6)	GNC 18–25 (n = 6)	BISEXUALS 18–25 (n = 5)	LESBIANS 45–64 (n = 5)
UTI ever diagnosed							
Yes	16 (44.4%)	5 (55.6%)	2 (40%)	3 (50%)	2 (33.3%)	1 (20%)	3 (60%)
I don’t know	2 (5.6)	0	1 (20)	0	0	0	0
During the past week…							
Leak urine?	8 (22.2)	0	4 (80)	0	1 (16.7)	0	3 (60)
Leak urine just after you had finished?	7 (19.4)	1 (11.1)	3 (60)	0	1 (16.7)	0	2 (40)
Leak urine in connection with a sudden need to rush to urinate?	5 (13.9)	1 (11.1)	1 (20)	0	0	0	3 (60)
Leak urine in connection with laughing, sneezing, or coughing?	5 (13.9)	3 (33.3)	1 (20)	0	0	0	1 (20)
Leak urine in connection with physical activities, such as exercising or lifting a heavy object?	5 (13.9)	1 (11.1)	2 (40)	0	1 (16.7)	0	1 (20)
All LUTS Experienced? (Yes)	25 (69.4)	8 (88.9)	5 (100)	3 (50)	4 (66.7)	2 (40)	3 (60)

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
