# Peer review of "Bladder Health Experiences, Perceptions and Knowledge of Sexual and Gender Minorities"

_ijerph, 2019, doi:10.3390/ijerph16173170_

Round 1

Reviewer 1 Report

The article is interesting, but the number of participants is small. I would suggest performing a sample calculation to see if you answer the question.

Reviewer 2 Report

Thank you for the opportunity to review “Bladder Health Experiences, Perceptions and Knowledge of Sexual Gender Minorities”. There are vast gaps in the literature on transgender health, particularly as it relates to restroom use and bladder health. This study makes a unique contribution to the literature. I believe this manuscript would be strengthened with revision. My suggestions for revision are as follows:

Include the “and” in Sexual and Gender Minorities. Sexual and gender minorities are two distinct, yet overlapping groups. “Sexual Gender Minorities” reads like gender minorities who are sexual in nature, which I doubt is the intention. The NIH SGM office and all researchers in public health that I am aware of use “Sexual and Gender Minorities.”

I would rethink the use of “GNC” to describe an identity. Gender non-conforming is increasingly being used to describe gender expression, not identity. Non-binary, or sometimes genderqueer, are more commonly used as gender identities.

Tables 2 and 3 provides a level of detail than seems necessary. In the text, the overall findings from these tables are described and I think that’s adequate for table content as well.

Table 2 brings up questions as to how you categorized the respondents. Presumably, the column categories should map on to the gender identity and sexual orientation categories. However, there are people categorized as trans men who do not identify as such. There are those categorized as GNC who identify as trans men. There may be some logical explanation for this, but it’s not explained in the text. I suggest explaining how you constructed the column categories and why they don’t match up in Table 2. However, my comment above suggest this is all a bit TMI anyway, so you may cut the columns altogether.

Line 128, I think you meant you exclude those who are identified as heterosexual and cisgender women. Not either/or. If not, then I don’t understand the exclusion criteria.

Line 129, transgender women are generally born with prostates, so this is not only an indicator of a cisgender man

Line 140, I do not know what it means to be “trained and grounded in SGM cultural norms.” Please give an example or two.

Line 234, perhaps give an intro lead-in to this long quote. Seems to arrive out of nowhere. The reader gets the gist eventually how these quotes will be used in the manuscript, but it’s initially jarring.

Lines 335-345, are the real names of FG participants being used in these quotes? If so, please note that you had permission to use them. If not, note that they are pseudonyms.

Line 359, incomplete heading

The emerging themes are largely from trans participants. What about the other participants? It’s a finding in an of itself if trans participants are the only ones who had any problems to discuss.

Line 420, I am confused as to how there are 18 trans individuals when the n’s in table 2 suggest there are only 11. Please explain this discrepancy.

The end notes do not seem to match up with the text. Perhaps the problem starts at #13, which seems to be part of #12.

Reviewer 3 Report

The paper is very interesting. People in general not aware this problem. Congratulations. The weak point is the number of people: 36, but this study can further other research.
